# Machine Learning of Multi-Modal Tumor Imaging Reveals Trajectories of Response to Precision Treatment

**DOI:** 10.3390/cancers15061751

**Published:** 2023-03-14

**Authors:** Nesrin Mansouri, Daniel Balvay, Omar Zenteno, Caterina Facchin, Thulaciga Yoganathan, Thomas Viel, Joaquin Lopez Herraiz, Bertrand Tavitian, Mailyn Pérez-Liva

**Affiliations:** 1INSERM, PARCC, Université Paris Cité, F-75015 Paris, France; 2Cancer Drug Research Laboratory, Department of Medicine, Division of Medical Oncology, The Research Institute of the McGill University Health Center (RI-MUHC), Montréal, QC H4A 3J1, Canada; 3Nuclear Physics Group and IPARCOS, Department of Structure of Matter, Thermal Physics and Electronics, CEI Moncloa, Universidad Complutense de Madrid, 28040 Madrid, Spain; 4Radiology Department, AP-HP, European Hospital Georges Pompidou, F-75015 Paris, France

**Keywords:** multi-modal imaging, paraganglioma, machine learning, hierarchical clustering, treatment response

## Abstract

**Simple Summary:**

In order to evaluate precision cancer therapies, it would be advantageous to measure at the same time their action on tumor growth and on the biological target of the therapy. New non-invasive hybrid imaging techniques allow access to multiple quantitative parameters. Here, we trained machine learning classifiers of features extracted from longitudinal in vivo co-registered metabolic, vascular and anatomical images in a mouse model of paraganglioma. We show that machine learning identifies ensembles of tumor states that correspond to stages of tumor evolution with or without anti-angiogenic treatment. These classifiers define individual trajectories of tumor progression and response to treatment, supporting the use of machine learning analysis of multiparametric imaging for the identification of response to anti-angiogenic treatment in this rodent model.

**Abstract:**

The standard assessment of response to cancer treatments is based on gross tumor characteristics, such as tumor size or glycolysis, which provide very indirect information about the effect of precision treatments on the pharmacological targets of tumors. Several advanced imaging modalities allow for the visualization of targeted tumor hallmarks. Descriptors extracted from these images can help establishing new classifications of precision treatment response. We propose a machine learning (ML) framework to analyze metabolic–anatomical–vascular imaging features from positron emission tomography, ultrafast Doppler, and computed tomography in a mouse model of paraganglioma undergoing anti-angiogenic treatment with sunitinib. Imaging features from the follow-up of sunitinib-treated (*n* = 8, imaged once-per-week/6-weeks) and sham-treated (*n* = 8, imaged once-per-week/3-weeks) mice groups were dimensionally reduced and analyzed with hierarchical clustering Analysis (HCA). The classes extracted from HCA were used with 10 ML classifiers to find a generalized tumor stage prediction model, which was validated with an independent dataset of sunitinib-treated mice. HCA provided three stages of treatment response that were validated using the best-performing ML classifier. The Gaussian naive Bayes classifier showed the best performance, with a training accuracy of 98.7 and an average area under curve of 100. Our results show that metabolic–anatomical–vascular markers allow defining treatment response trajectories that reflect the efficacy of an anti-angiogenic drug on the tumor target hallmark.

## 1. Introduction

Establishing treatment response is a crucial aspect of precision oncology [1]. This determination involves categorizing the patient’s status into predefined discrete classes [2,3]. These classes are established by pooled assessment of the margins of variation of descriptive features extracted from medical data or images. Given the wide existing variety of medical data, imaging systems, and clinical protocols in oncology, there are standardized recommendations for defining treatment response. The World Health Organization (WHO) in 1979 determined four categories of response or non-response to treatment based on tumor volume [4]: a complete, partial, stable, and progressive disease. In 2000, the Response Evaluation Criteria for Solid Tumors (RECIST) proposed to sum one-dimensional measurements of the greatest length of all lesions extracted from X-ray tomography (CT) or magnetic resonance imaging (MRI) images [5]. The RECIST criteria have periodically been revised, and new versions have emerged to accommodate new targeted therapies. In 2009, the Positron Emission Tomography (PET) Response Criteria for Solid Tumours (PERCIST) was introduced to provide a continuous variable for categorizing patient response to treatment. This involves calculating the percentage change between pre-and post-treatment PET scans of the peak standard uptake value (SUL) corrected for body mass or the sum of all SULs of all lesions. The RECIST and PERCIST criteria provide classification labels that respond to the macroscopic characteristics of tumors and are robust and convenient for clinical practice. However, they provide little, if any, information about the effect of a precision treatment on its pharmacological target, e.g., immune checkpoint inhibition, anti-angiogenesis, and targeted immunotherapy. Therefore, RECIST and PERCIST are of limited interest for the evaluation of new treatments, which contrasts with the increasing availability of in vivo molecular and functional imaging approaches targeting tumor hallmarks [6], and even the interaction of these hallmarks through hybrid imaging [7,8,9]. Thus, new tumor response criteria specific to the pharmacological target being addressed are needed.

Artificial intelligence (AI), a term derived from the informatics field, has shown promising potential to accelerate the evolution of healthcare toward precision oncology [3,10]. In particular, machine learning (ML), a branch of AI that applies statistical methods to detect patterns within datasets, enables the assembly and analysis of large volumes of data and facilitates diagnosis, prognosis, and treatment response assessment [3,10,11,12,13,14,15,16,17,18]. Traditionally, unsupervised ML clustering methods have been used to cluster the molecular and/or genomics patient profiles and to analyze in response to treatment [18] with posterior supervised learning generalization [19]. These early “omics” studies have laid the groundwork for more recent analyses using profiles created with radiology imaging features, known as radiomics. Radiomics provide a large number of quantitative features that can be used by ML methods to detect high-dimensional patterns that correlate with relevant clinical endpoints. Because they can be applied to routinely acquired images at no cost, radiomics have expanded to almost all branches of molecular imaging [20,21,22], anatomical imaging [23,24,25] and hybrid imaging [12]. However, radiomics techniques possess several limitations. Firstly, the biological significance of the imaging features extracted through radiomics is often unclear. To overcome this limitation, certain studies have attempted to establish correlations between radiomics features and manually crafted biological descriptors derived from the images [17]. However, numerous radiomics features remain inadequately understood and their clinical applicability is hampered by a lack of interpretability [26,27]. Secondly, radiomics involves a vast number of features computed using predefined mathematical expressions [12]. Given that translational research datasets are often limited in size, it is probable that employing numerous features may result in overfitting during machine learning (ML) training [28]. Therefore, most radiomics studies concentrate on large clinical databases. On the other hand, preclinical studies, which, due to animal experimentation regulations, rely on small databases, often favor the use of a few handcrafted clinical image descriptors with direct biological interpretation [29].

In this study, we investigate the response to an antitumoral treatment of paragangliomas (PGLs), rare neuroendocrine tumors arising from extra-adrenal chromaffin cells that originate from the neural crest cells and are characterized by high metabolism and extensive vascularization [30]. Sunitinib is an anti-angiogenic drug used to treat patients with PGLs [31]. In previous work by our team, we showed that the response to sunitinib treatment in experimental PGLs-bearing mice was highly variable [32]. In some animals, the tumors responded well to sunitinib, while in other animals, the tumors resumed growth in just a couple of weeks [32]. During treatment, we documented the vascular (using ultrafast Doppler imaging (UDI)), metabolic (using PET), and anatomical (using CT) responses of mice to sunitinib using a new hybrid imaging system that combines PET-registered ultrafast sonography (PETRUS) [33]. Imaging with PETRUS sunitinib-treated or sham-treated mice documented the effect of sunitinib on tumor growth, vessels development, and 2${}^{\prime }$-[${}^{18}$F]fluoro-2${}^{\prime }$deoxy-D-glucose (FDG) uptake [32].

Here, we combine hierarchical clustering analysis (HCA) and supervised ML classifiers to identify different stages of tumor progression and the response of PGLs undergoing sunitinib or sham treatments using a few longitudinal-handcrafted vascular–molecular–anatomical features with direct biological interpretation. Multiple classical ML classifiers exist with simplified models suitable for small preclinical databases such as ours, and to date, it has not been explored which classifier is best suited to the task of identifying response to the sunitinib treatment of PGL using multimodal descriptors. Therefore, in this work, we evaluated several ML classifiers and used the one with the best performance for the generalized classification of tumor progression stages. The concatenation of the resulting stages along the duration of anti-angiogenic or sham treatments resulted in the identification of trajectories of tumor evolution.

## 2. Materials and Methods

Figure 1 shows the pipeline of the framework implemented in this study that progresses from the acquisition of multi-modal image volumes to the definition of individual trajectories of response to treatment. Each element of this diagram will be described in the following sections.

### 2.1. Acquisition of Live Animal IMAGING Data

Two groups of mice followed the protocol of animal housing, tumor implantation, follow-up, and anti-angiogenic drug delivery described in [32] and schematized in Figure 2. The first group (training group) included 16 mice from [32], while the second group (validation group) included another 11 mice that underwent the same experimental protocol. Imaging of the training group was performed at baseline, and then every week until week 3 for vehicle-treated animals (8 mice), and every week until week 6 for sunitinib-treated animals (8 mice). The validation group concerned only sunitinib-treated animals, and imaging was performed at the baseline, week 1, week 3 and week 6.

Animal experiments were approved by the French Ethical committee under reference No. 16-098 and performed by certified personnel following the French law on animal experimentation n°2013-118. In brief, adult female nude 6-week-old mice weighing 30 g (Janvier Labs, France) were implanted in the dorsal fat pad with tumors obtained from immortalized mouse chromaffin cells (imCC) carrying a homozygous knockout of the Sdhb gene (Sdhb${}^{-/-}$) as previously described [32]. Mice were housed under controlled temperature (24 °C), relative humidity (50%), a 12/12 light/dark cycle, and free access to water and food. When the tumor volume reached 140 mm${}^{3}$, mice were randomly divided into a vehicle group (CON, *n* = 8) and a sunitinib group (SUNI, *n* = 8). The sunitinib group received sunitinib malate (Clinisciences, A10880-500) daily at a dose of 50 mg/kg body weight for 6 consecutive weeks, administered by oral gavage of 200 µL in a 10 mg/mL DMSO/PBS (1:4) solution. The control group received daily 200 μL doses of the DMSO-PBS solution (1:4) for 3 weeks. Mice were euthanized if the tumor volume exceeded UKCCCR recommendations [34] or if they showed signs of advanced cancer disease.

The effect of sunitinib was monitored non-invasively using the hybrid In vivo imaging technology PETRUS (positron emission tomography registered ultrafast sonography) [33], which allows for the simultaneous acquisition of tissue metabolism using [${}^{18}$F]Fluorodeoxyglucose (FDG) PET, computed tomography (CT) and ultrafast ultrasound Doppler imaging (UUDI) [33]. PETRUS simultaneously reads the cellular metabolism activity alongside the micro-vascular architecture within the tumor, ensuring unimpaired physiological conditions for both sets of spatially co-registered features [32].

### 2.2. Description of Database Formation

Each PETRUS acquisition comprised three image volumes registered in a common time and space reference frame that defined a multiparametric cube surrounding the animal tumor. The features describing the metabolic, vascular, and anatomical characteristics of the tumor were extracted from the PET, UUDI, and CT images, respectively (Table 1). A volume of interest (VOI) covering the whole tumor was defined on the PET images by segmenting voxels with an FDG standard uptake value (SUV) greater than 30% of the tumor’s peak SUV at 50–60 min post-injection [35]. This VOI was used to create a binary mask that was applied to the three spatiotemporal registered volumes. From the masked PET image, the following metabolic features were extracted: mean, coefficient of variance, minimum and maximum of standard uptake values (MeanSUV, CVstdSUV, MinSUV, MaxSuv), and PET volume (PETVolume). The masked UUDI volume was filtered using a Hessian-based vessel enhancement filter, and vessels were segmented using predefined thresholds [36] and skeletonized using an iterative ordered thinning-based skeletonization method [37,38]. The skeletonized mask of vessels was transformed into a graph of nodes and edges representing the vascular network of the tumor. Using this graph, the following features describing the topology of the tumor vascularization were calculated: mean, minimum and maximum vessel length (MeanVesselsLength, MinVesselsLength, MaxVesselsLength), mean vessels tortuosity (Tort), which is the shortest distance between nodes divided by the vessel length), vessels length dispersion (VesselsLength-Disp), which is the standard deviation of the vessels length divided by the mean of the vessels length, number of nodes (NumNodes), density of nodes (DensityNodesinUSV), mean vessels diameter (MeanVesselsDiam) and ultrasound volume (USVolume), which is the number of voxels of the vascular skeleton multiplied by the voxel volume. The quantification of PETRUS images was performed using MATLAB version R2021b. The CT volume (CTVolume) was delineated from the fat pad surrounding the tumor.

The working database assembled all 15 features extracted from the imaging modalities, as well as a unique record number that defined the mouse, the week of the imaging session (where week zero (W0) is the pre-treatment imaging session and W1-6 is the rest of the treatment weeks), and the treatment group assignment (CON for sham-treated mice; SUNI for sunitinib-treated mice). Data were divided into 3 subgroups, (i) $D_{training}^{suni}$ containing the SUNI mice in the training group, aggregating a total of 54 records (ii) $D_{training}^{con}$ containing the CON mice from the training group, forming a total of 27 records, and (iii) $D_{validat}^{suni}$ containing the SUNI mice from the validation group forming a total of 28 records.

### 2.3. Feature Selection

Feature selection is an important pre-processing step that affects the accuracy and decreases the training time of any classifier. By removing non-useful or redundant features, the dimensionality of the feature space can be reduced, an essential step to improve the performance of a classifier [39]. In order to identify linear correlations between the different features, we applied a Pearson correlation using a Pearson coefficient |r|> 0.9 (*p*-value < 0.05) to detect redundant features [40]. In addition, non-informative features with a low coefficient of variation (CV < 0.1) were removed.

### 2.4. Unsupervised Classification: Hierarchical Clustering

One of the fundamental objectives of our study was the determination of phenotypically representative clusters, each cluster being a representative combination of metabolic, anatomical and vascular features associated with a stage of response to sunitinib. Clusters were determined by the individual response of the subject, independently of the time of treatment by assembling all the longitudinal features extracted. HCA, an unsupervised machine-learning clustering approach [41], was used to stratify the tumor response by finding common metabolic, anatomical and vascular phenotypic patterns of the image descriptors selected. The HCA was applied on each of the training datasets separately, $D_{training}^{suni}$ and $D_{training}^{con}$, in order to determine whether or not the treatment changes the time course of tumor evolution. First, the input data were standardized using the z-score. Then, the interrelationship between individual records was measured by computing the unweighted average Euclidean distance. This was followed by computing the average link as a similarity metric to define the closest pair of clusters. Finally, a heat map with dendrograms was constructed to display the patterns observed and the clusters identified. The length of the dendrogram branches connecting records and features is inversely proportional to the similarity of their profiles. Gap statistics [42] was applied in order to evaluate the optimal number of clusters, and Welch’s *t*-test was applied to identify significantly different clusters [43]. The outcome of this analysis provided the optimal number of clusters corresponding to a particular phenotype identified for each instance in the data-base. HCA and statistical tests were implemented in MATLAB (version 2021-b) using the clustergram, ttest2, and evalclusters functions, respectively.

### 2.5. Supervised Classification: Model Building and Validation

To test the stability of the method, we compared the clustering results applied on an external population ( $D_{validation}^{suni}$) to a classification produced as a generalization of the clustering performed on our initial population ($D_{training}^{suni}$). More precisely, we considered the clusters of the initial population ($D_{training}^{suni}$) as classes of a supervised classification algorithm to predict the classes expected in the new population ($D_{validation}^{suni}$).

Because our training dataset has an unbalanced number of instances per class, which can undermine the predictability of the models, we performed oversampling through the synthetic minority over-sampling technique (SMOTE), which balances the minority classes [44]. This technique uses the k-nearest neighbors approach to synthesize new observations based on the existing records. We applied smote using the four nearest neighbors to balance each of the four clusters (A, B1, B2, and C).

The selected features of our $D_{training}^{suni}$ were brought into ten machine learning classifiers, including decision tree (DT), Gaussian naive Bayes (GNB), kernel naive Bayes (KNB), linear support vector machine (Linear SVM), quadratic support vector machine (Quadratic SVM), k-nearest neighbors (KNN), weighted k-nearest neighbors (Weighted KNN), random forest (RF), narrow neural network (Narrow NN), bilayered neural network (Bilayered NN). The best-performing model was selected by comparing the area under the receiver operating characteristic curve (AUC) and accuracy (ACC) values. The control parameters of the best model were further optimized by Bayesian optimization and five-fold cross-validation to evaluate the performance of the classifier. All classifiers were trained and validated using the *classification learner* application implemented in MATLAB version 2021-b.

In order to check the relative importance of each of the metabolic, vascular, and morphological features in the classification problem, we used the predictor importance attribute associated with the RF model. The predictor importance attribute is an implicit technique performed using the RF model and is evaluated using the Gini impurity criterion index. This index is based on the principle of impurity reduction to provide the power of each feature in the classification [45].

### 2.6. Identification of Trajectories of Treatment Responses

We then tested whether the records assembled within each cluster, corresponding to a tumor state with specific biomarkers, could represent a chronological stage of tumor evolution. By referring back to the time point of each record (the week after the beginning of treatment) in both the CON and SUNI groups, the clusters were ordered chronologically, and a time-dependent trajectory was obtained for each mouse. We applied an $R^{2}$ test to the states at each of the seven time points of the study (classes obtained from the HCA, considering A = 1, B1 = 2, B2 = 3, and C = 4) to determine if these states indicated temporal stages of treatment response. Finally, the transitional matrix between clusters was analyzed.

## 3. Results

### 3.1. Pearson Correlation

Figure 3 shows the cross-heatmap of the Pearson correlation values (r) of CT, vascular, and metabolic features. In order to eliminate redundant features, a Pearson significance of r > 0.9 and *p*-value < 0.05 were applied to all pairs of features of the four instances. This reduced the number of vascular features from 11 to 8: MeanVesselsLength was correlated with MeanVesselsDiameter, Tort, and VesselsLengthDisp; VesselsLengthDisp correlated with MeanVesselsDiameter and Tort, and Tort correlated with MeanVesselsDiameter. Hence, MeanVesselsLength, MeanVesselsDiameter and Tort were not considered further. Applying the same Pearson r and *p* values reduced the metabolic features from 5 to 4: MeanSuv correlated with MaxSuv, and MaxSuv was not considered further.

With respect to vascular–metabolic correlations, interestingly, the StdSUV was significantly correlated with MeanVesselsDiam and MeanVesselsLength.

In addition, a low coefficient of variation (CV < 0.1) results in a non-informative dataset from classifiers’ training. Thus, features having a high Pearson correlation and a low coefficient of variation were not considered further. Overall, 8 features, including 4 vascular features, i.e., USVolume, NumNodes, DensityNodesinUSV, VesselsLengthDisp, 3 metabolic features, i.e., StdSUV, PETVolume, MeanSUV, and the CT volume, were used for all three curated databases ($D_{training}^{suni}$, $D_{training}^{con}$, $D_{validat}^{suni}$).

### 3.2. Hierarchical Clustering Approach

#### 3.2.1. Sham-Treated Training Set ($D_{training}^{con}$)

Performing the hierarchical clustering on the $D_{training}^{con}$ dataset identified two major clusters: Clusters $A_{c}$ and $C_{c}$ (Figure 4a), where subscript *c* stands for the control group. They showed the following characteristics (Table 2):Cluster $A_{c}$ was characterized by significantly low volumes of CT, PET, and UUDI, a high coefficient variance of the standard deviation of SUV, a low number of nodes, and a low density of nodes. This corresponds to a small-sized tumor, with low vascularization and metabolism, and a heterogeneous distribution of FDG uptake.Cluster $C_{c}$ was characterized by high volumes of CT, PET, and UUDI, a significantly lower coefficient of variation of the standard deviation of SUV, and a high number of nodes. This cluster corresponds to a stage where the tumor has grown to a large volume, with high metabolic and vascularization activities but a low heterogeneity in the distribution of FDG uptake.

#### 3.2.2. Sunitinib-Treated Training Set ($D_{training}^{suni}$)

The same clustering approach applied to the $D_{training}^{suni}$ dataset identified three major clusters (Figure 4b): Clusters $A_{t}$, $B_{t}$, and $C_{t}$, where the subscript *t* stands for the treatment group. Cluster $B_{t}$ splitted into two subgroups: $B1_{t}$ and $B2_{t}$ (Table 3).

Cluster $A_{t}$ was characterized by low volumes of CT, PET, and UUDI, a high coefficient of variation of the standard deviation of SUV, and low vessel length dispersion, number of nodes, and density of nodes. This corresponds to a small-sized tumor with low vascularization and heterogeneous distribution of FDG uptake value, features that are similar to those of cluster $A_{c}$ of the control group.Cluster $C_{t}$ was characterized by high volumes of CT, PET, and UUDI, low coefficient of variation of the standard deviation of SUV, high vessel length dispersion, and very high number of nodes. This cluster corresponds to a tumor with a large volume, high metabolism and vascularization, and low heterogeneity in the distribution of FDG uptake, features that are similar to those of cluster $C_{c}$ of the control group.

To compare the *A* and *C* clusters obtained with the SUNI and CON groups, respectively, a Kruskal–Wallis test [46] was performed between the $A_{t}$ and $A_{c}$ clusters, and also between the $C_{t}$ and $C_{c}$ clusters. The clusters were statistically similar (*p*-value < 0.05), indicating that clusters $A_{t}$ and $A_{c}$ on the one hand, and clusters $C_{t}$ and $C_{c}$ on the other hand, correspond to similar tumor states in the sunitinib-treated and sham-treated groups.

In the sunitinib-treated training set, the HCA algorithm identified two further clusters not present in the CON group:Cluster $B1_{t}$ was characterized by low to moderate volumes of CT, PET, and UUDI, low coefficient of variation of the standard deviation of the SUV, high vessel length dispersion, and a very high density of nodes. This corresponds to a small tumor with a significant but moderate level of vascularization, and medium-to-high heterogeneity in the distribution of FDG uptake.Cluster $B2_{t}$ was characterized by moderate volumes of CT and PET, high UUDI volume, lower coefficients of variation of the standard deviation of SUV, high vessel length dispersion, and low density of nodes. This corresponds to a moderate to high tumor volume and vascularization and low heterogeneity in the distribution of FDG uptake.

### 3.3. Robustness of Clusterization

An additional validation step was performed in order to ascertain that cluster formation was reproducible and not a casuistic process. HCA clustering was repeated on subsets of random instances of the $D_{training}^{suni}$ group, formed by randomly removing one mouse at a time. The accuracy of each HCA was calculated by considering the clusters obtained for all mice as ground truth and comparing it with the clusters of the new subset using the following formula: $Accuracy=\frac{Number\;ofcorrectpredictions}{TotalnumberofPredictions}$. As shown in Table 4, the total accuracy for each of the performed HCAs was greater than 95 percent for the three major clusters ($A_{t}$, $B_{t}$, and $C_{t}$).

### 3.4. Performance of Supervised Machine Learning Models

All 10 of the ML classifiers explored demonstrated good predictive performance, as demonstrated by the evaluation indexes of performance presented in Figure 5a. GNB achieved the best predictive performance (AUC: 100, ACC: 98.7), whereas DT exhibited the weakest (AUC: 96, ACC: 94.8). The remaining classifiers achieved the following predicted performance: Quadratic SVM (AUC: 100, ACC: 97.4), KNB (AUC: 98, ACC: 94.8), Linear SVM (AUC: 100, ACC: 97.4), KNN (AUC: 97, ACC: 98.7), RF (AUC: 100, ACC: 94.8), Narrow NN (AUC: 100, ACC: 96.1), Bilayered NN (AUC: 98, ACC: 94.8) and Weighted NN (AUC: 100, ACC: 97.4).

Applying the best classifier to the three records that had not been classified using HCA, i.e., mouse 1-week 6, mouse 3-week 6, and mouse 8-week 5, allowed to classify these records into clusters $C_{t}$, $C_{t}$, and $A_{t}$, respectively (Table 5). This classification remained consistent with the previous stages of the sunitinib training set $D_{training}^{suni}$. The best-trained model applied to the $D_{validat}^{suni}$ dataset assigned a state for each record and mouse (Table 6) that was consistent with the states of the $D_{training}^{suni}$ dataset.

Finally, using the RF classifier the relative importance of features used for training showed that all three types of tumor features, i.e., metabolic, vascular, and anatomical features, participated in the prediction of the four clusters (Figure 5b). This indicates that the information provided by each of the three imaging modalities contributed in a balanced way to define tumor stages for each imaging record.

#### Clusterization Reveals Tumor Progression

We then tested whether the different clusters would correspond to different time points during the tumor follow-up, i.e., whether, for any record, there was a correlation between assignment to one particular cluster and the time point at which imaging had been performed for that record. Regarding the CON group, all except two records (mouse 3/week 2 and mouse 6/week 2) of cluster $A_{c}$ corresponded to the baseline or to the week-1 time point. Conversely, all cluster $C_{c}$ records corresponded to week-2 or week-3 acquisitions. This confirms that cluster $A_{c}$ represents an initial stage of the tumor, while cluster $C_{c}$ represents an advanced tumor stage.

In contrast, the correspondence between the time-point of acquisition and assignment to the $A_{t}$ or $C_{t}$ cluster was much looser for the SUNI group than for the CON group. For example, mouse 6 remained in cluster $A_{t}$ at all time points until week 6. Moreover, at baseline and week 1, a significant number of mice were not assigned to the $A_{t}$ cluster but either to the $B1_{t}$ cluster (two mice at baseline and three at week 1) or to the $B2_{t}$ cluster (two at each time point). Conversely, upon reaching the last observation time point (week 6), five mice from the SUNI group were in the $C_{t}$ cluster, while one was classified in the $B1_{t}$ cluster, one in the $B2_{t}$ cluster, and one in the $A_{t}$ cluster. Examples of trajectories for a mouse from the sham-treated group and for two mice from the sunitinib-treated group are shown in Figure 6a.

We then investigated the influence of the vascular and metabolic features on the clustering results. Removing PET and UUDI features from the SUNI datasets and basing clustering only on the CT volume led to the co-clustering of [$A_{t}$;$B1_{t}$] and [$B1_{t}$;$B2_{t}$] (see boxplot in Figure 7a). This indicates that RECIST-like criteria using only CT did not identify intermediate clusters. When the same algorithm HCA was applied to the SUNI dataset from which the vascular features obtained by ultrasound imaging had been removed, i.e., using only the PET metabolic features and the CT volume, only two significantly different clusters were obtained using gap statistics: clusters $A_{PET/CT}$ and $B_{PET/CT}$. This indicates that PERCIST-like criteria, using PET-CT only, did not identify intermediate clusters (Figure 7b). Therefore, the intermediate B stage ($B_{t}$) and its two sub-clusters $B1_{t}$ and $B2_{t}$, essentially reflect changes concerning the vascular features of tumors under sunitinib treatment.

### 3.5. Clusters Depict Responses to Sunitinib Treatment

To further understand how clusters reflect the response to sunitinib treatment, the evolutionary trajectories (passage from one cluster to another over successive time points) were studied individually for each mouse of the SUNI group (Table 5). The progression from cluster $A_{t}$ to $C_{t}$ of sunitinib-treated mice was not direct as the $A_{c}$ to $C_{c}$ in the sham-treated animals but passed through intermediate $B_{t}$ clusters. This was confirmed by a correlation analysis performed on clusters $A_{t}$, $B_{t}$ and $C_{t}$ considered stages 1, 2 and 3, resulting in $R^{2}$ = 0.84. Calculation of the cluster transition matrix confirmed the relationship between the clusters and the chronology of tumor evolution. Assuming a progression represented by states $A_{t}$, $B_{t}$, and finally $C_{t}$, we obtained 29/46 (65.9%) stable phenotypes, i.e., remaining in the same state; 10/46 (22.7%) one progression, i.e., advancing further to the next state; and 5 (11.3%) regressions from $B_{t}$ to $A_{t}$ (Appendix A
Table A1). Pooling the validation population and the training population showed an asymmetry between “progression” (*n* = 15) and “regression” (*n* = 6). Finally, Cluster $C_{t}$ was an irreversible transition deriving essentially from the $B2_{t}$ state that appeared as a mandatory intermediate stage to reach state $C_{t}$, and the transition from $B_{t}$ to $A_{t}$ occurred only by the intermediary stage $B1_{t}$, and not by $B2_{t}$. States $A_{t}$, $B1_{t}$, $B2_{t}$, and $C_{t}$ are thus ordered in time, suggesting that they are in fact tumor stages and that there is a progressive evolution of tumor stages from states $A_{t}$ to $C_{t}$ through $B_{t}$, and irreversibly between $C_{t}$ and the other states.

In summary, multi-feature ML analysis of the sunitinib-treated animals showed that individual trajectories, defined by the passage from one cluster to another, followed a discrete number of rules:Irrespective of whether mice received sunitinib or vehicle, no mouse reversed from the advanced tumor stage (cluster *C*) to a less advanced stage.In the sunitinib group, mice moved from the early tumor stage (cluster $A_{t}$) to either one of the two intermediate stages (clusters $B1_{t}$ or $B2_{t}$) but not directly to the advanced stage (cluster $C_{t}$).In the sunitinib group, mice moved from cluster $B1_{t}$ to $B2_{t}$ and back, and from cluster $B1_{t}$ back to cluster $A_{t}$, but no passage from cluster $B2_{t}$ to cluster $A_{t}$ was observed.In the sunitinib group, all mice reaching the advanced (cluster $C_{t}$) stage originated from cluster $B2_{t}$.

The robust correlations between clusters and treatment duration, and the transition matrix between clusters confirm that the A, B, and C clusters correspond to tumor stages. Interestingly, transitions between sub-clusters $B1_{t}$ and $B2_{t}$ were less correlated with time than transitions between $A_{t}$ and $B1_{t}$ or $B2_{t}$, and between $B2_{t}$ and $C_{t}$. This suggests that the “reverse” transitions, i.e., $B2_{t}$ to $B1_{t}$ and $B1_{t}$ to $A_{t}$, could reflect the phenotype changes associated with a positive response to sunitinib. Figure 8 summarizes the trajectories between tumor stages in sunitinib-treated mice. There was first an increase in the level of tumor vascularization ($A_{t}$ to $B1_{t}$ transformation), followed by a decrease in the heterogeneity of FDG distribution in the tumor ($B1_{t}$ to $B2_{t}$).

## 4. Discussion

Previous studies used ML to study the correspondence between gene expression and tumor progression [47,48], including PGL [49]. To the best of our knowledge, this is the first application of ML based on HCA and supervised ML algorithms to noninvasive multimodal imaging of PGL. PGL lesions may concern the whole sympathetic and parasympathetic chains from the base of the skull to the pelvis. Germline mutations in one of the SDHx genes are responsible for approximately 20% of cases of PGL and also in some other tumors [50,51]. PGL patients carrying SDHx mutations show a higher rate of metastatic disease and a lower rate of survival than non-SDHx PGL patients. Surgery is not without risk and may be impractical for numerous or misplaced lesions. Clinical trials with sunitinib have reported modest results in SDHB mutation carriers [32,52].

There is an international consensus on the use of repeated non-invasive imaging for the screening, management and follow-up of PGL patients [53], as well as for asymptomatic SDHx mutation carriers [54]. Our results show that unsupervised ML of serial noninvasive and multimodal imaging data can define the phenotypic stages of mouse Sdhb${}^{-/-}$ PGL tumors under anti-angiogenic treatment. The main finding is that, although the records fed to the ML algorithm had not been time stamped for the duration of treatment, unsupervised ML applied to multimodal multiparametric imaging features yielded clusters relevant to disease progression and to the response to sunitinib. In the sham-treated group, all mice switched, generally in less than three weeks, from cluster $A_{c}$, an early stage with small and poorly developed tumors, low vascularization, and heterogeneous FDG uptake, to cluster $C_{c}$, an advanced stage with large tumors, large vessels, high and relatively homogeneous FDG uptake, corresponding to an end-stage cancer disease. In the sunitinib-treated group, a given tumor from a given mouse could, over time, move from one cluster to another, suggesting that the changes from one cluster to another depicted trajectories of tumor evolution related to the response or the escape from treatment. Some sunitinib-treated tumors showed a progression similar to sham-treated tumors, which infers that sunitinib-treated mice entering the advanced-stage $C_{t}$ cluster have escaped sunitinib treatment.

Two other clusters, $B1_{t}$, and $B2_{t}$, representing intermediate tumor stages, were observed only in the sunitinib-treated group, supporting the view that their phenotypes represent the effects of sunitinib on PGL tumors. The first one, $B1_{t}$, encompassed small-sized tumors with a significant but moderate level of vascularization and heterogeneity in the distribution of glucose uptake. The second cluster, $B2_{t}$, encompassed tumors of moderate volume and vascularization, and low heterogeneity in the distribution of glucose uptake. ML did not identify these two intermediate stages when the vascular features derived from ultrafast ultrasound were removed from the analysis. Therefore, the $B1_{t}$ and $B2_{t}$ intermediate stages identified the effect of sunitinib on tumor vascularization, likely by inhibition of vascular endothelial growth factors receptors (VEGFRs), the major pharmacological target of the drug [55]. Previous studies have documented the relationship between tumor vascular types and the malignancy of PGL or pheochromocytoma, which is the adrenal form of paraganglioma. In a pioneering study, Favier et al. [56] divided pheochromocytomas into two groups according to their vascular architecture. Tumors with short, straight vascular segments distributed regularly over large areas of tumoral tissue had a vascular density equivalent to that observed in the normal adrenal medulla, while tumors with longer vascular segments of irregular length and a lower density of vessels corresponded to the malignant form. These regular and irregular patterns observed using in vitro stained sections of tumor tissue samples are remarkably similar to the states that we observed here in vivo, A and C [56]. A few years later, a study attempted to use “Favier’s criteria” of the vascular patterns on histological sections of pheochromocytomas and PGL for the prediction of clinical behavior [57]. Again, malignancy was associated with an irregular vascular pattern; however, in spite of the correct agreement between observers, sensitivity and specificity were relatively modest and the authors concluded that vascular patterns, although useful, were not sufficient as “stand-alone […] prognostic tool for the distinction between benign and malignant PCC…”. Interestingly, we observed a difference in vascular morphology reminiscent of regular/irregular patterns under sunitinib treatment, tumor vessels being larger in diameter at stage $B2_{t}$ than at stage $B1_{t}$ (see Figure 6b). Therefore, while the analysis of vascularization may by itself not be sufficient, and notwithstanding the fact that the morphology of vessels in fixed tissue may not reflect their in vivo morphology, there is good agreement with changes in vessel morphology and the response to sunitinib, suggesting that the in vivo exploration of vascular morphology may be useful for the management of PGL. In addition, the link between FDG heterogeneity and microvascular density was theorized using a spatiotemporal computational model [58]. Our present results are in agreement with the authors’ conclusion that “as microvascular densities increase […], the spatiotemporal distribution of total FDG uptake by tumor tissue changes towards a more homogenous distribution [58]”. Therefore, combined imaging of vascularization and metabolism could be an advantage for the follow-up of PGL patients under treatment.

Interestingly, all of the three mice that pertained to a *B* cluster ($B1_{t}$ or $B2_{t}$) at baseline ended up in the $C_{t}$ cluster at the end of the 6-week sunitinib treatment, while only one of the four mice pertaining to the $A_{t}$ cluster at baseline ended up in the $C_{t}$ cluster. Although further studies are necessary to determine whether the tumor’s biology prior to the administration of sunitinib could predict future escape from treatment, this may indicate that tumors that have already developed a significant vessel network are less prone to respond to sunitinib therapy. Thus, even though the switch from $B1_{t}$ to $B2_{t}$ was reversible under sunitinib treatment ($B1_{t}$ to $B2_{t}$), increased vascularization and decreased metabolic heterogeneity defining the $B2_{t}$ stage were necessary features for passage to the $C_{t}$ stage, in other words, for escape from sunitinib treatment. From a cancer biology point of view, this suggests that escape from sunitinib treatment involves both a metabolic and a vascular switch.

From a statistical point of view, the analysis of each record independently without time stamping allows to extraction of information regarding the rates of tumor evolution in a small group of eight mice. This would not have been possible with conventional methods based on time-stamped groups of individuals unless the number of individuals would have been drastically increased. Considering the necessity to reduce the use of animals in research, the unsupervised method for the analysis of multimodal imaging presented here is an attractive alternative for the preclinical exploration of treatments in cancer models.

Moreover, cluster extraction using multiple features could allow gaining a better understanding of the sequence of events underlying drug response. The fact that cancer is a multiform disease with multiple intermingled hallmarks has been extensively documented and reviewed in the classical paper by Hanahan and Weinberg [59]. Therefore, it is unlikely that assessing only one biomarker, even one that informs on the activity toward the pharmacological target, may be sufficient to assess treatment response, and, even less so, to identify complex escape mechanisms. All in all, our results support the recourse to multimodal imaging with the careful selection of relevant imaging biomarkers, ideally including one or several biomarker(s) of the hallmark targeted by the treatment. In this respect, other tumor variants could also benefit from similar approaches extracting biomarkers specific to the tumor type and/or treatment. Finally, it may also be interesting to apply a radiomics analysis in order to compile mathematically defined image features and determine whether they represent phenotypic states predictive of tumor stage predictive of treatment response.

The main limitation of our study is that it is based on preclinical data. Serial imaging sessions, even non-invasive, are difficult to envision in clinical settings. However, we show that comprehensive longitudinal explorations in a patient-relevant animal model can identify key imaging features leading to sunitinib resistance, and may inspire translational methods for tumor follow-up in patients. ML analysis of multimodal hybrid imaging could offer individual monitoring of the vascular and metabolic states of a tumor, thus providing valuable information for personalized treatment decisions. Our results need to be further validated on prospective cohorts and extended to the clinical situation.

## 5. Conclusions

The combination of hierarchical clustering and supervised machine learning algorithms provides remarkable insight into the progression of tumor development in a mouse model of paraganglioma. Through the incorporation of multi-modal information, including the vascular features of the tumor-targeted by sunitinib, our approach is successful in depicting trajectories of response to treatment. This approach could set a basis for personalized follow-up of tumors treated by targeted therapies. 

## Figures and Tables

**Figure 1 cancers-15-01751-f001:**
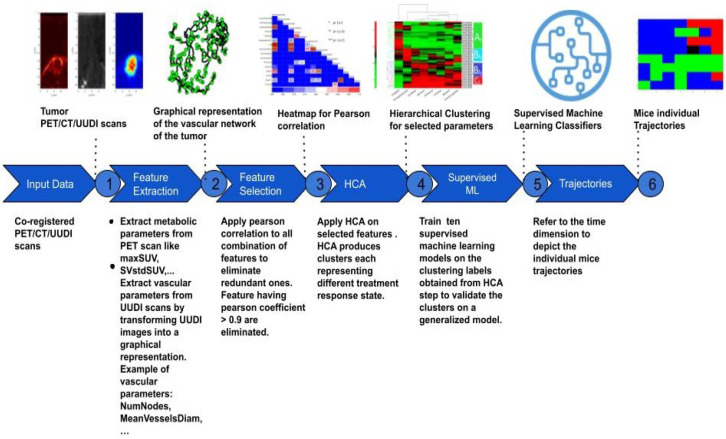
Process diagram showing the framework pipeline. Images were co-registered and processed to extract features describing the metabolic, vascular, and morphological components of tumor development. A Pearson correlation study was performed to remove redundant features. Longitudinal features were combined, and hierarchical clustering analysis was applied to obtain clusters and classes representing different stages of tumor evolution. The clusters and classes identified with HCA were used with 10 different supervised machine-learning classifiers for model generalization and final validation. Finally, time-wise concatenation of the identified stages was performed to form the individual trajectories of tumor evolution for each animal.

**Figure 2 cancers-15-01751-f002:**
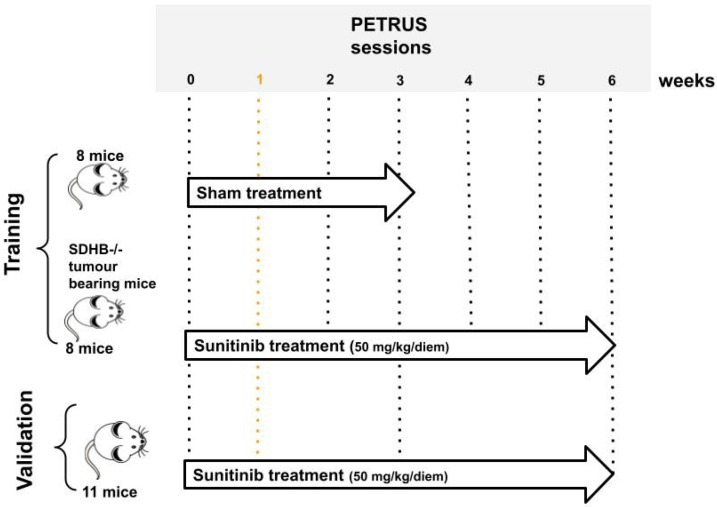
Database generation process. Mice in the training group were divided into two groups: sunitinib-treated and sham-treated. Eight mice from each group were scanned with PETRUS before and after 1, 2, and 3 weeks of treatment. Sinitinib-treated mice were also imaged at 4, 5, and 6 weeks of treatment. Mice of the independent validation set were sunitinib-treated and scanned at baseline and at weeks: 1, 3, and 6 of the treatment.

**Figure 3 cancers-15-01751-f003:**
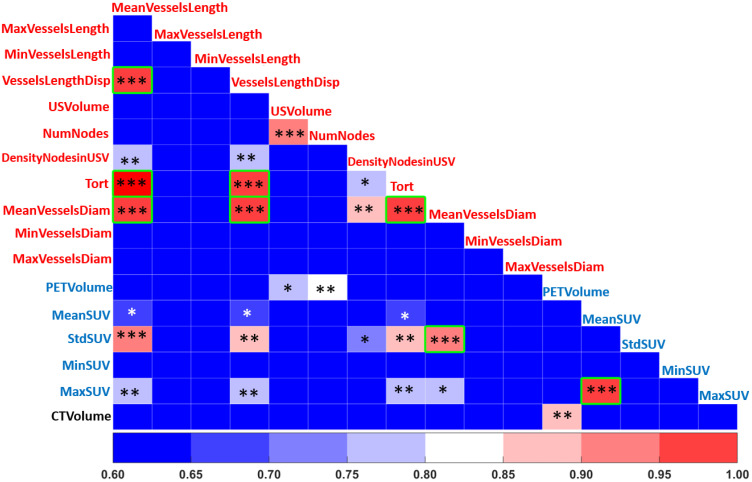
Heatmap summarizing significant Pearson coefficient values for each pair of metabolic (blue font), vascular (red font) and morphological features (black font) used to exclude redundant features (*, **, ***, refer to *p*-value level of significance).

**Figure 4 cancers-15-01751-f004:**
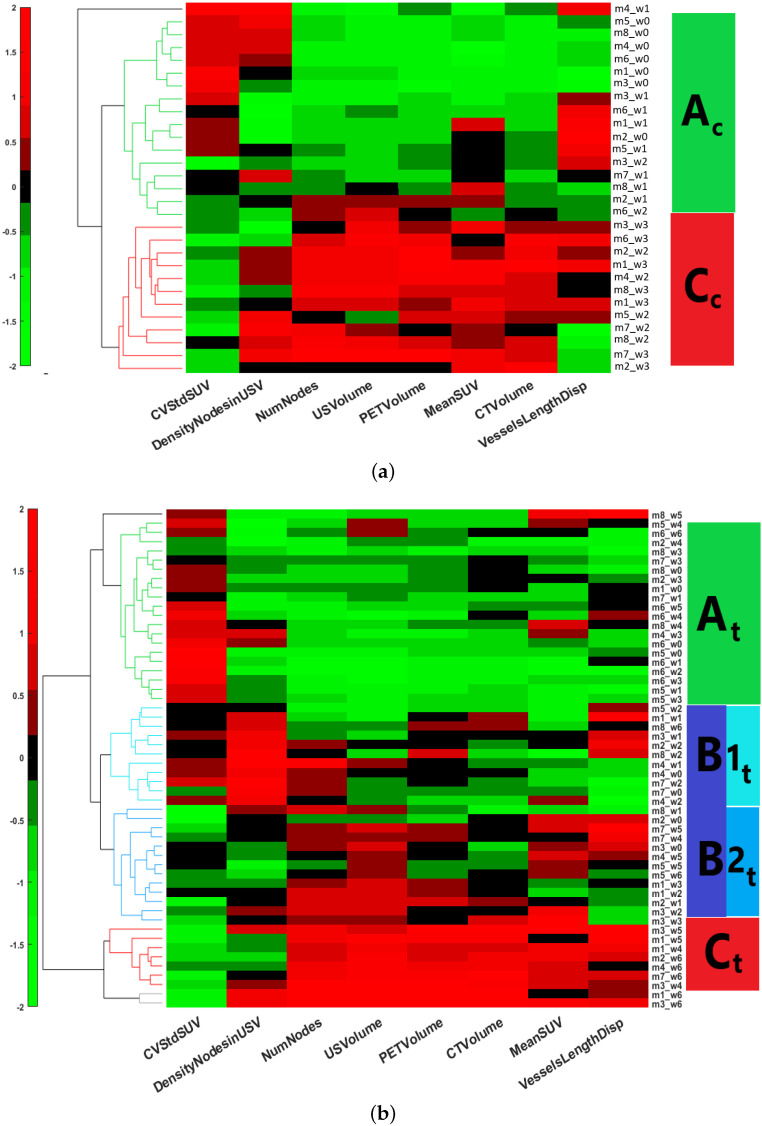
Heatmap and hierarchical clustering performed (**a**) on the $D_{training}^{con}$ dataset and (**b**) on the $D_{training}^{suni}$ dataset. Two clusters ($A_{c}$, $C_{c}$) were identified in (**a**) and 4 clusters ($A_{t}$, $B1_{t}$, $B2_{t}$, and $C_{t}$) were identified in (**b**).

**Figure 5 cancers-15-01751-f005:**
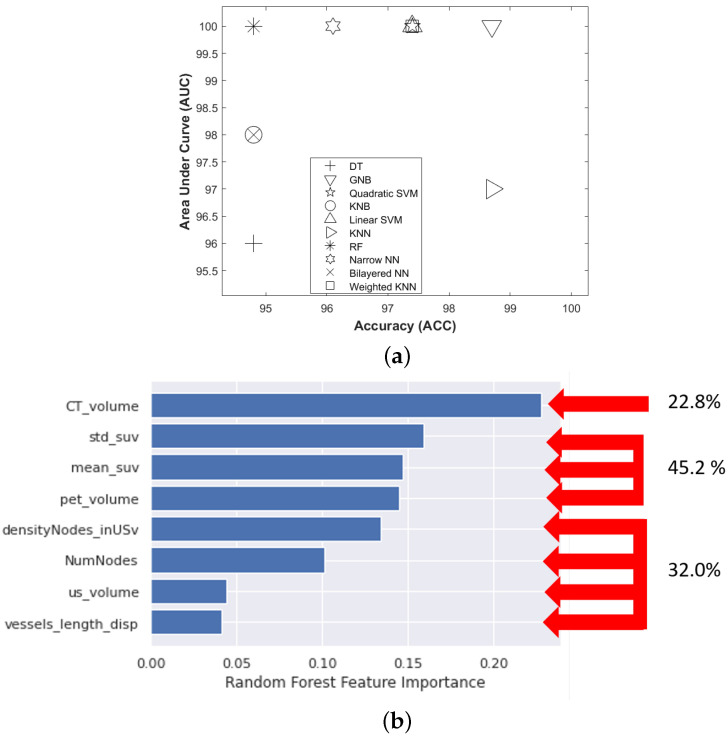
Performance of the supervised machine learning models (**a**) Scatter diagram of machine learning classifiers prediction performance. The horizontal axis represents accuracy (ACC), the vertical axis represents the area under the curve (AUC); DT, decision tree; GNB, Gaussian naive Bayes (Gaussian); Quadratic SVM, support vector machine (Quadratic); KNB, kernel naive Bayes; Linear SVM, linear support vector machine; KNN, k-nearest neighbors; RF, random forest; NNN, narrow neural network; Bilayered NN, bilayered neural network; Weighted KNN, weighted k-nearest neighbors. (**b**) Contribution of morphological, metabolic, and vascular features in the discrimination of the 4 clusters of tumor evolution stages identified with RF.

**Figure 6 cancers-15-01751-f006:**
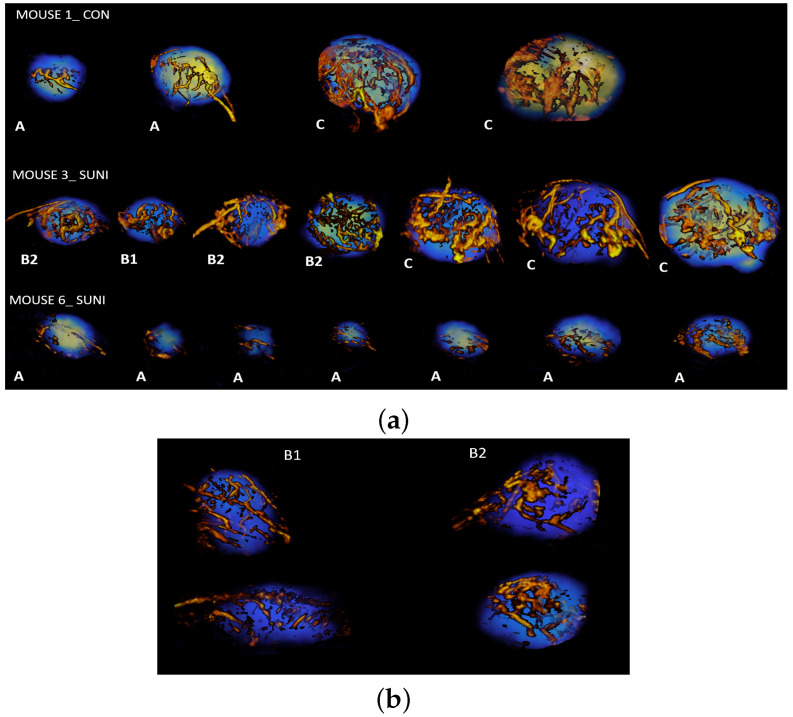
Maximum intensity projection renderings (MIP) of PGL tumors, (**a**) mouse 1 from the CON group, mouse 3 and mouse 6 from the SUNI group. Tumors in the CON group are shown at baseline and from week 1 to week 3, while tumors from the SUNI group are shown at baseline and at week 1 to week 6. (**b**) Comparison of PGL tumors at the $B1_{t}$ and $B2_{t}$ stages.

**Figure 7 cancers-15-01751-f007:**
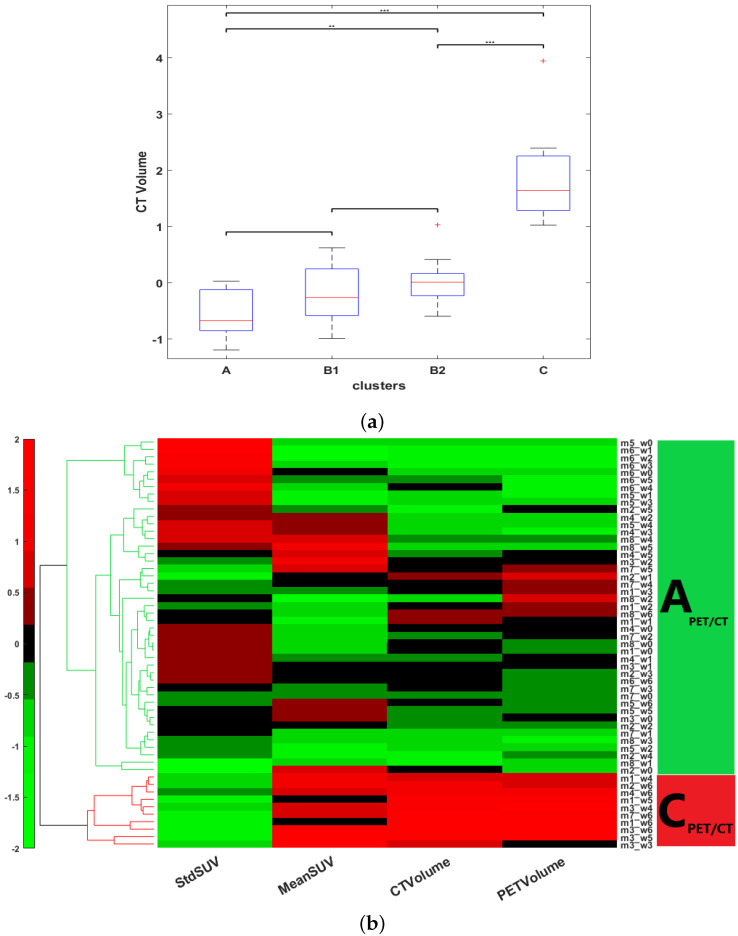
Contribution of the vascular features for cluster discrimination in the SUNI group (**a**) CTVolume shows no significant difference between $A_{t}$-$B1_{t}$ and $B1_{t}$-$B2_{t}$ (*p*_value > 0.05), indicating that RECIST criteria alone did not identify the intermediate B1 and B2 clusters. (**b**) Similarly, hierarchical clustering performed on the $D_{training}^{suni}$ dataset considering only the features derived from PET and CT scans did not identify the intermediate stages $B1_{t}$ and $B2_{t}$ either.

**Figure 8 cancers-15-01751-f008:**
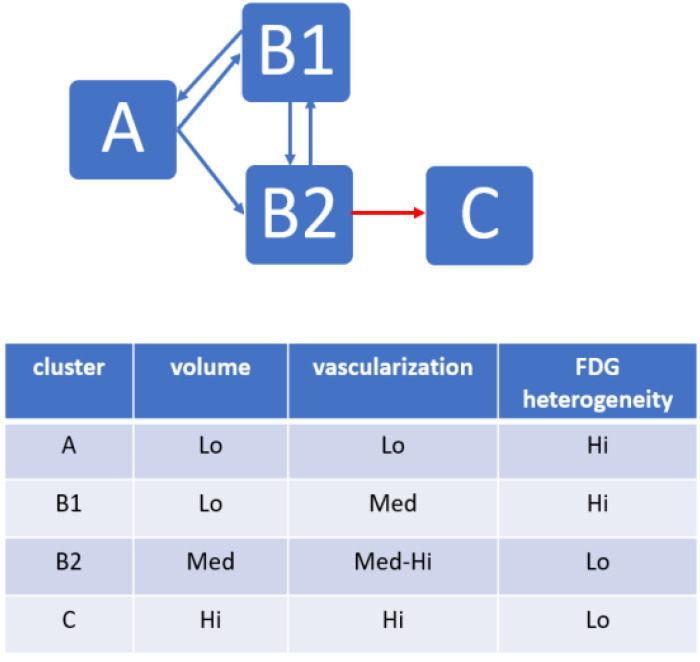
Graphical and tabular representations of the trajectories highlighting the major characteristic features of mice under sunitinib treatment.

**Table 1 cancers-15-01751-t001:** PET/CT/UUDI extracted features.

Parameter	Modality	Abbreviation	Unit	Description
Mean Standardized Uptake Value	PET	Mean SUV	a.u.	Average of the Standardized Uptake of FDG in the VOI
Max Standardized Uptake Value	PET	Max SUV	a.u.	Average of the 5 hottest pixels in the tumor VOI
Min Standardized Uptake Value	PET	Min SUV	a.u.	Minimum Standard Uptake of FDG in the VOI
Standardized Uptake Value of FDG dispersion	PET	CVstdSUV	a.u.	Coefficient of variance of the Standardized Uptake Value
PET volume	PET	PETvolume	${\mathrm{mm}}^{3}$	Number of voxels in the VOI × volume of a voxel
Computed Tomography Volume	CT	CTVolume	${\mathrm{mm}}^{3}$	Tumor volume defined by the CT scan
Number of Nodes	UUDI	NumNodes	nodes	Sum of all Nodes.
Number of Nodes/Vessels Volume	UUDI	DensityNodesinUSV	nodes/ ${\mathrm{mm}}^{3}$	Number of nodes per unit of vessel volume.
Maximum Vessels Length	UUDI	MaxVesselsLength	mm	Average of the maximum length of all the vessels
Mean Vessels Length	UUDI	MeanVesselsLength	mm	Average of the length of all the vessels
Minimum Vessels Length	UUDI	MinVesselsLength	mm	Average of the min length of all the vessels
Length Vessels Dispersion	UUDI	VesselsLengthDisp	a.u.	Coefficient of variance of the mean vessel length
Mean Vessels Tortuosity	UUDI	Tort	a.u.	Average of all tortuosities. The tortuosity is the ratio between the length of a vessel (as an arc) and the straight-line length between its initial and final points
Mean Vessels Diameter	UUDI	MeanVesselsDiam	mm	Average of all mean Diameter
Vessels Volume	UUDI	USVolume	${\mathrm{mm}}^{3}$	Tumor blood volume defined by the Ultrasound Doppler scan

**Table 2 cancers-15-01751-t002:** Metabolic, vascular, and morphological characteristics of the clusters of the $D_{training}^{con}$ dataset. The average values of each parameter of each cluster are represented. In black, the mean values; in parenthesis, the standard mean errors; and in blue, the z-score means.

Features	CVstd-SUV	Density Nodes inUSV (1/mm${}^{3}$)	Num-Nodes	US Volume (mm${}^{3}$)	PET Volume (mm${}^{3}$)	CT Volume (mm${}^{3}$)	Mean SUV	Vessels Length Disp (mm${}^{2}$)
Cluster $A_{c}$	45.07 (1.68), 0.42	36.27 (1.85), −0.27	542.85 (32.84), −0.81	15.31 (1.02), 0.82	236.43 (27.75), −0.85	165.06 (23.80), −0.85	1.96 (0.10), −0.63	60.06 (1.89), 0.00
Cluster $C_{c}$	35.74 (0.89), 0.42	38.79 (1.57), −0.27	1549.29 (123.44), −0.81	39.44 (2.50), 0.82	815.28 (69.49), −0.85	584.29 (51.77), −0.85	2.66 (0.08), −0.63	59.60 (1.19), 0.00

**Table 3 cancers-15-01751-t003:** Metabolic, vascular, and morphological characteristics of the clusters from the $D_{training}^{suni}$ dataset. The mean values of each parameter of each cluster are represented. In black, the means; in parentheses, the standard means error; and in blue, the z-score means.

Features	CVstd-SUV	Density Nodes inUSV (1/mm${}^{3}$)	Num-Nodes	US Volume (mm${}^{3}$)	PET Volume (mm${}^{3}$)	Mean SUV	CT Volume (mm${}^{3}$)	Vessels Length Disp (mm${}^{2}$)
Cluster $A_{t}$	52.01 (1.17), 0.81	28.99 (1.05), −0.64	243.15 (18.20), −0.90	5.58 (0.78), −0.81	99.61 (9.73), −0.73	1.73 (0.11), −0.56	66.90 (8.22), −0.60	55.68 (0.77), −0.52
Cluster $B1_{t}$	47.68 (0.65), 0.12	44.08 (1.84), 1.46	527.4 (50.99), 0.16	11.84 (0.80), −0.34	195.85 (18.44), −0.09	1.79 (0.15), −0.49	100.08 (15.36), −0.29	57.33 (2.22), −0.24
Cluster $B2_{t}$	43.93 (0.72), −0.64	32.06 (1.02), −0.07	583.64 (29.78), 0.61	18.11 (0.46), 0.78	228,16 (15.76), 0.50	2.56 (0.22), −0.35	123.51 (11.52), 0.06	59.46 (1.40), −0.25
Cluster $C_{t}$	41.13 (0.58), −0.92	31.78 (1.29), −0.25	790.67 (39.46), 1.13	24.89 (0.73), 1.52	386.45 (29.66), 1.17	2.67 (0.12), 0.60	261.73 (17.84), 1.23	63.93 (2.01), 0.89

**Table 4 cancers-15-01751-t004:** Performance of each of the HCAs for subsets of the $D_{training}^{suni}$ dataset. Data subsets were obtained by removing all the time points of one mice at a time.

Mice Removed	1	2	3	4	5	6	7	8
Total Accuracy (%)	100	100	95	98	100	100	95	100

**Table 5 cancers-15-01751-t005:** Evolutionary path of sunitinib-treated mice of the training set. Items marked as * indicate missing classification due to the absence of corresponding PETRUS data. Clusters that were assigned by the RF model are underlined.

Mouse Number	Baseline	Week 1	Week 2	Week 3	Week 4	Week 5	Week 6
mouse 1	** $A_{t}$ **	** $B1_{t}$ **	** $B2_{t}$ **	** $B2_{t}$ **	** $C_{t}$ **	** $C_{t}$ **	** $C_{t}$ **
mouse 2	** $B2_{t}$ **	** $B2_{t}$ **	** $B1_{t}$ **	** $A_{t}$ **	** $A_{t}$ **	*	** $C_{t}$ **
mouse 3	** $B2_{t}$ **	** $B1_{t}$ **	** $B2_{t}$ **	** $B2_{t}$ **	** $C_{t}$ **	** $C_{t}$ **	** $C_{t}$ **
mouse 4	** $B1_{t}$ **	** $B1_{t}$ **	** $B1_{t}$ **	** $A_{t}$ **	*	** $B2_{t}$ **	** $C_{t}$ **
mouse 5	** $A_{t}$ **	** $A_{t}$ **	** $A_{t}$ **	** $A_{t}$ **	** $A_{t}$ **	** $B2_{t}$ **	** $B2_{t}$ **
mouse 6	** $A_{t}$ **	** $A_{t}$ **	** $A_{t}$ **	** $A_{t}$ **	** $A_{t}$ **	** $A_{t}$ **	** $A_{t}$ **
mouse 7	** $B1_{t}$ **	** $A_{t}$ **	** $B1_{t}$ **	** $A_{t}$ **	** $B2_{t}$ **	** $B2_{t}$ **	** $C_{t}$ **
mouse 8	** $A_{t}$ **	** $B2_{t}$ **	** $B1_{t}$ **	** $A_{t}$ **	** $A_{t}$ **	** $C_{t}$ **	** $B1_{t}$ **

**Table 6 cancers-15-01751-t006:** Clusterization of the 11 sunitinib mice from the validation group. Items marked as - indicate that the RF approach was unable to assign the record to one any of the $A_{t}$, $B1_{t}$, $B2_{t}$, $C_{t}$ clusters. Items marked as * indicate no PETRUS data available.

Mouse Number	Baseline	Week 1	Week 3	Week 6
mouse 9	** $A_{t}$ **	** $B1_{t}$ **	** $B1_{t}$ **	*
mouse 10	** $A_{t}$ **	** $A_{t}$ **	** $B1_{t}$ **	** $C_{t}$ **
mouse 11	** $A_{t}$ **	** $B1_{t}$ **	** $B2_{t}$ **	*
mouse 12	** $A_{t}$ **	** $B1_{t}$ **	** $C_{t}$ **	-
mouse 13	** $A_{t}$ **	** $A_{t}$ **	*	** $C_{t}$ **
mouse 14	** $A_{t}$ **	** $B2_{t}$ **	*	*
mouse 15	** $B1_{t}$ **	** $B1_{t}$ **	*	*
mouse 16	** $B1_{t}$ **	** $B1_{t}$ **	*	*
mouse 17	** $A_{t}$ **	** $A_{t}$ **	** $B1_{t}$ **	*
mouse 18	** $B1_{t}$ **	** $A_{t}$ **	*	*
mouse 19	** $A_{t}$ **	*	*	*

## Data Availability

The data presented in this study are available in this article.

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
