# Peer review of "Machine Learning of Multi-Modal Tumor Imaging Reveals Trajectories of Response to Precision Treatment"

_cancers, 2023, doi:10.3390/cancers15061751_

Round 1
Reviewer 1 Report
The study presents a novel machine learning-based framework for the analysis of multi-modal imaging data in a mouse model of paraganglioma under sunitinib treatment. The authors aim to overcome the limitations of traditional methods that assess the response to cancer treatments based on gross characteristics such as tumor size or glycolysis.
The framework was tested on a dataset of both vehicle and sunitinib-treated animals and was validated on an independent dataset of sunitinib-treated mice. The results showed that unsupervised hierarchical clustering was effective in classifying multiple image-derived features into defined treatment response stages. The application of a Random Forest classification model to validate the clusters and discriminate between different stages of treatment response was also successful.
The conclusion of this study highlights the potential of addressing high dimensional problems with unsupervised machine learning. The ability to define trajectories of response to treatment based on the efficacy of sunitinib on its target tumor hallmark is a valuable contribution to the field and has the potential to inform future treatment decisions and improve patient outcomes.
Overall, this study provides a promising new approach to the analysis of multi-modal imaging data in cancer treatment and shows the potential for machine learning to inform clinical decision making.
The manuscript can be improved from the following aspects:
1. The results and conclusion in the abstract are unclear to readers. The numerical result and conclusion from the study should be reported clearly.
2. Line 48. Recently, machine learning techniques that go beyond tumor size measurement have been proposed to refine cancer diagnosis. The prior literature should be discussed separately and introduce the rational connections with the current study.
3. Line 132. Feature selection is an important pre-processing step that affects the learning rate and quality of any classifier. The learning rate is a hyperparameter and predefined. The feature selection cannot impact the learning rate.
Author Response
"Please see the attachment."

Reviewer 2 Report
Line 2- such as shoul dlist out the characteristics
Instead of going with multiple assessment platforms like matlab and python for different methods it is advisable to stick on with one platform.
State of art literature survey is preferrable
Eventhough the work seems to include more efforts, the Radom forest algorithm usage seems to be of little less effort in th eML based analysis.
Multiple ML methods or hyperparameter optimisation coul dbe involved to show an innovativeness in the work taken up. The initial parts of the article like data collection, dataset description has involved much effort but the sections covering ML shoul dbe innovative enough to take up a comparision with the state of art methods.
